# Longitudinal assessment of neuronal 3D genomes in mouse prefrontal cortex

Amanda C. Mitchell[1,2], Behnam Javidfar[1], Lucy K. Bicks[1,2], Rachael Neve[3], Krassimira Garbett[4], Sharon S. Lander[5], Karoly Mirnics[4], Hirofumi Morishita[1,2], Marcelo A. Wood[6], Yan Jiang[1,2], Inna Gaisler-Salomon[5] & Schahram Akbarian[1,2]

Neuronal epigenomes, including chromosomal loopings moving distal *cis*-regulatory elements into proximity of target genes, could serve as molecular proxy linking present-day-behaviour to past exposures. However, longitudinal assessment of chromatin state is challenging, because conventional chromosome conformation capture assays essentially provide single snapshots at a given time point, thus reflecting genome organization at the time of brain harvest and therefore are non-informative about the past. Here we introduce 'NeuroDam' to assess epigenome status retrospectively. Short-term expression of the bacterial DNA adenine methyltransferase Dam, tethered to the *Gad1* gene promoter in mouse prefrontal cortex neurons, results in stable G$^{methyl}$ATC tags at *Gad1*-bound chromosomal contacts. We show by NeuroDam that mice with defective cognition 4 months after pharmacological NMDA receptor blockade already were affected by disrupted chromosomal conformations shortly after drug exposure. Retrospective profiling of neuronal epigenomes is likely to illuminate epigenetic determinants of normal and diseased brain development in longitudinal context.

[1] Department of Psychiatry, Friedman Brain Institute, Icahn School of Medicine at Mount Sinai, 1470 Madison Avenue, New York, New York 10029, USA. [2] Department of Neuroscience, Friedman Brain Institute, Icahn School of Medicine at Mount Sinai, 1470 Madison Avenue, New York, New York 10029, USA. [3] McGovern Institute for Brain Research, Massachusetts Institute of Technology, 43 Vassar Street, Cambridge, Massachusetts 02139, USA. [4] Department of Psychiatry, Vanderbilt University, 1211 Medical Center Drive, Nashville, Tennessee 37232, USA. [5] Department of Psychology, University of Haifa, 199 Aba Khoushy Avenue Mount Carmel, 3498838 Haifa, Israel. [6] Department of Neurobiology and Behavior, University of California at Irvine, 2205 McGaugh Hall, Irvine California 92697, USA. Correspondence and requests for materials should be addressed to S.A. (email: Schahram.akbarian@mssm.edu).

A large number of genetic and environmental factors impacting within the extended (prenatal to young adult) period of brain development result in cognitive and behavioural deficits only at much later periods. Unsurprisingly, therefore, considerable time and effort has been invested exploring long-term adaptations of neuronal and glial transcriptomes and epigenomes in a wide range of disease models[1–4]. However, functional neurogenomics faces a key limitation, owing to the fact that, to date, even the most powerful genome-scale assays for chromatin modifications (chromatin immunoprecipitation (ChIP) sequencing), transcription (RNA sequencing) or chromosomal conformations (Hi-C) essentially provide only a single snapshot of genome organization and function at the time of tissue harvest. This often necessitates multiple subgroups of animals, to monitor, at different time points, long-term genomic adaptations in response to past exposure. What is missing from the field, therefore, is a molecular toolbox that would directly link past genome status to current brain function. Here we artificially tag neuronal genomes in mouse prefrontal cortex (PFC) with bacterial DNA adenine methyltransferase (Dam), then measure the animal's behaviour at different points in time, followed by brain harvest and retrospective assessment of prefrontal genomes representing the exposure period months past. Specifically, we show that chronic deficits in cognition and working memory, and excessive anxiety after 21 days of disrupted NMDA (N-methyl-D-aspartate) receptor signalling in the juvenile and young adult period are associated with early emergence of long-lasting disruptions of intra-chromosomal conformations at the NMDA-sensitive Gad1 GABA synthesis gene locus (chr.2qC2). We predict that in vivo Dam-based retrospective tagging of neuronal genomes (hereafter referred to as 'NeuroDam') will provide an important longitudinal complement to conventional cross-sectional neuroepigenomic approaches currently available.

## Results

**NMDA antagonist-induced long term behavioural defects.** Transient disruption of NMDA receptor signalling induces lasting impairments in neuronal signalling, cognition, social behaviours and emotion, and provides a frequently implied pharmacological model for schizophrenia and other psychosis spectrum disorder[5,6]. Specifically, acute or subchronic ($<$21 days) exposure to MK-801 and other NMDA receptor antagonist drugs in the juvenile or young adult period is associated with disruptions of cortical function and cognition including associate and working memory, and increased anxiety[7–10]. To recapitulate these findings and to explore potential long-term changes in behaviour, we exposed two age groups of C57Bl6/J mice (postnatal day P28 and P90) to daily treatments with the NMDA antagonist drug MK-801 (0.2 mg kg$^{-1}$) or saline as control for a period of 3 weeks, followed by behavioural testing for spatial working memory (8-arm radial maze) and anxiety (open-field test) within 2 weeks ('TIME A' in Supplementary Fig. 1) or after 4 months ('TIME B' in Supplementary Fig. 1) after the last drug treatment. Indeed, MK-801-exposed mice showed significant deficits in working memory and increased anxiety when tested within 2 weeks post treatment ('TIME A') (Supplementary Fig. 1). These alterations continued to exist, in milder form, at the second, much later test period ('TIME B') (Supplementary Fig. 1). Although our findings are in broad agreement with studies using shorter time intervals between NMDA antagonist treatment and behavioural assessment[7–10], the results from our animals tested 4 months after MK-801 exposure indicate that such types of behavioural alterations do not remain static in the long term.

**Dam-tagging of chromosomal contacts in longitudinal context.** NMDA receptor blockade induces promoter-specific DNA methylation remodelling in corticolimbic circuitry[11], but little is known about potential effects on higher-order chromatin, including chromosomal conformations bypassing linear genome to mobilize enhancers and other cis-regulatory elements into physical proximity to (NMDA sensitive) gene transcription start sites (TSSs). We wanted to chart, in the longitudinal context of the aforementioned drug-induced changes in behaviour, chromosomal loop-bound DNA sequences encompassing Gad1/Gad67, an activity-regulated gene highly sensitive to disruptions in NMDA receptor signalling[12,13]. Given the protracted course of behavioural deficits after NMDA blockade, potentially emerging and extending over the course of multiple weeks, we reasoned that cross-sectional chromosome conformation capture (3C) approaches are less ideal to fully capture the dynamics of spatial genome architectures in susceptible neurons. We noted a report of bacterial adenine DNA methyltransferase (Dam)-based tagging of long-range chromosomal loopings at the Drosophila bithorax homeotic gene complex with chimeric Dam-GAL4 DNA binding domain constructs[14]. Therefore, we asked whether it would be possible to artificially tag Gad1-bound chromosomal contacts in mouse brain expressing chimeric protein comprising Dam fused to designer DNA-binding proteins targeting Gad1 promoter sequences. We hypothesized that such type of approach in vivo is well suited for long-term tagging of neuronal DNA because of postmitotic status, with the potential for a stable artificial DNA mark in the absence of the 'diluting effect' by cell division. Furthermore, vertebrate genomes essentially lack endogenous adenine methylation at G$^m$ATC tetramers as the highly specific Dam target sequence[15].

To test this 'NeuroDam' approach, we first designed a herpes simplex vector (HSV) amplicon for simultaneous expression of two transcription cassettes, arranged in tandem nose-to-tail orientation, including cytomegalovirus promoter driven mCherry (or green fluorescent protein (GFP)) and HSV IE4/5 promoter-driven expression of Dam fused to a transcription activator-like effector (TALE) complementary DNA previously shown to specifically bind to the predicted 14 bp target sequence at the Glutamic acid decarboxylase (Gad1, chromosome 2qC2) promoter[16]. We reasoned that HSV amplicons are ideal vectors for the purposes of NeuroDam retrospective genomics, because expression is rapid starting 2–3 h after transfection, but confined to a short period of several days before shutting down permanently[17,18]. Indeed, immunohistochemical staining with NeuN neuronal antibody confirmed expression of our HSV TALE$^{Gad1}$Dam/mCherry amplicon in neuronal layers of the cerebral cortex at day 2 (Fig. 1a,c) but not day 10 post injection (Fig. 1b). Furthermore, when tested by quantitative reverse transcriptase–PCR with two independent primer pairs ('Dam.1' and 'Dam.2' in Fig. 1d), prefrontal Dam RNA was readily detectable in brains harvested 2 and 7, but not 10 days post injection. Furthermore, RNA levels of Gad1 and its paralogue, Gad2, remained unaltered when tested 2, 7 and 10 days post injection (Fig. 1d and Supplementary Fig. 2). We conclude that HSV-mediated TALE$^{Gad1}$Dam expression is transient and does not alter RNA levels of the target gene.

Mice, treated for 3 weeks with daily doses of saline or MK-801 as described above, received—after an additional 2-week interval—bilateral PFC injections of HSV TALE$^{Gad1}$Dam/mCherry ('week 5' on Fig. 2 timeline). Brains were harvested after another 2 weeks post injection, on completion of 'TIME A' behaviour testing ('week 7' in Fig. 2), or after 4 months post injection, on completion of 'TIME B' behavioural assays (Supplementary Fig. 1) ('week 22' on Fig. 2 timeline). We first explored whether artificial, Dam-mediated adenine methylation

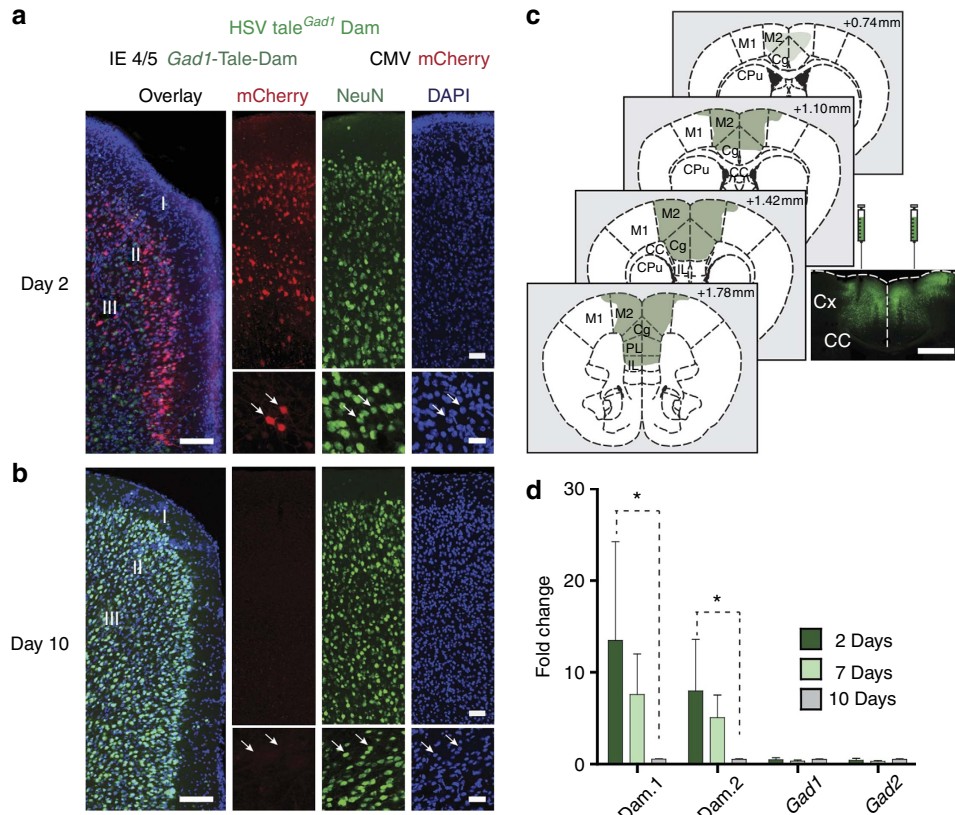

**Figure 1 | Transient Dam expression in PFC neurons.** (**a,b**) Tissue sections from PFC harvested (**a**) 2 and (**b**) 10 days after HSV-TALE$^{Gad1}$Dam/mCherry injection. Notice robust mCherry (red) expression in neuronal layers at day 2 but not day 10 post injection. Section were immunolabelled for NeuN neuronal marker (green) and counterstained with 4,6-diamidino-2-phenylindole (blue). (**a,b**) Scale bars, 100 μm (left); 50 μm (upper right); 25 μm (lower right). (**c**) Representative example of bilateral HSV-TALE$^{Gad1}$Dam/GFP injection, showing robust transgene expression within ∼1 mm rostrocaudal strip of rostro-medial cortex. Microscopical image from Bregma + 1.42 mm injection site, showing bilateral needle tracks and robust GFP expression in surrounding tissue. Scale bar, 1 mm. CC, corpus callosum; Cg, cingulate cortex; CPu, caudate putamen; Cx, cortex; M1, M2, motor cortex; PL (IL), pre-(infra-)limbic cortex. (**d**) Quantitative reverse transcriptase–PCR to quantify *Dam* and, for comparison, *Gad1* and *Gad2* RNA expression 2, 7 and 10 days after injection of HSV amplicon encoding TALE$^{Gad1}$Dam. Dam RNA, assayed with two independent primer pairs Dam.1 and Dam.2, expressed as fold change after normalization to 18S rRNA. $N = 4$ per time point; data shown as mean ± s.e.m. Notice highest Dam expression on day 2. *$P = 0.029$, Mann–Whitney test (both Dam.1 and Dam.2).

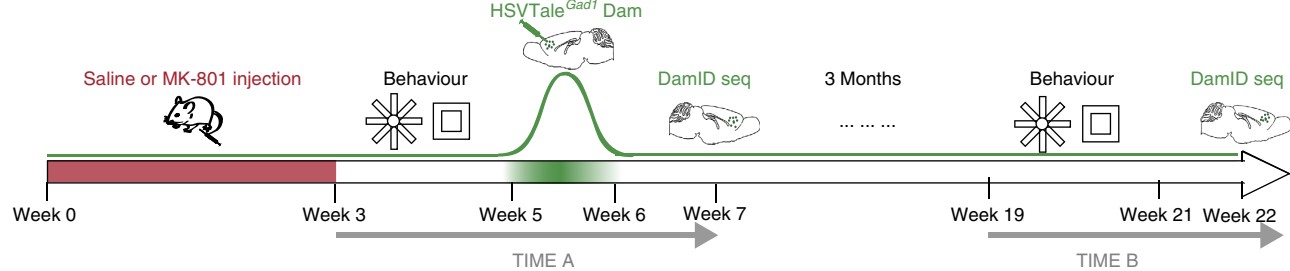

**Figure 2 | Experimental timeline.** Juvenile (P28) and young adult (P90) animals receive from week 0–3 daily injections of MK-801 or saline, followed by radial arm maze and open-field behavioural assays. At the beginning of week 5, HSV TALE$^{Gad1}$Dam vector was injected bilaterally into PFC. Brains from 'TIME A' mice were harvested at the end of week 6/beginning of week 7. Brains from 'TIME B' mice were harvested in week 22 and after another round of behavioural testing had been completed within week 19–21.

at GATC tetramers (G$^m$ATC) could identify some of the previous reported higher-order chromatin structures, based on conventional chromosome conformation capture assays (3C)[19]. These included loop contacts between the *Gad1* promoter and regulatory sequences positioned 55 kb further upstream[19]. We extracted, then DpnII digested the prefrontal DNA, followed by PCR-based quantification of restriction-insensitive G$^m$ATC sequences indicative of Dam methylation activity[20] (Fig. 3a,b).

Indeed, quantification of DpnII-resistant sequences within 100 kb from the *Gad1* TALE target site identified in multiple experiments a sharp peak corresponding to the previously reported conformation[19] (Fig. 3c). To further assess the sequence specificity of our TALE$^{Gad1}$Dam DNA-binding protein (which included a V5 epitope tag), we conducted chromatin immunoprecipitation with an anti-V5 antibody in additional PFC samples, harvested 2 days after HSV

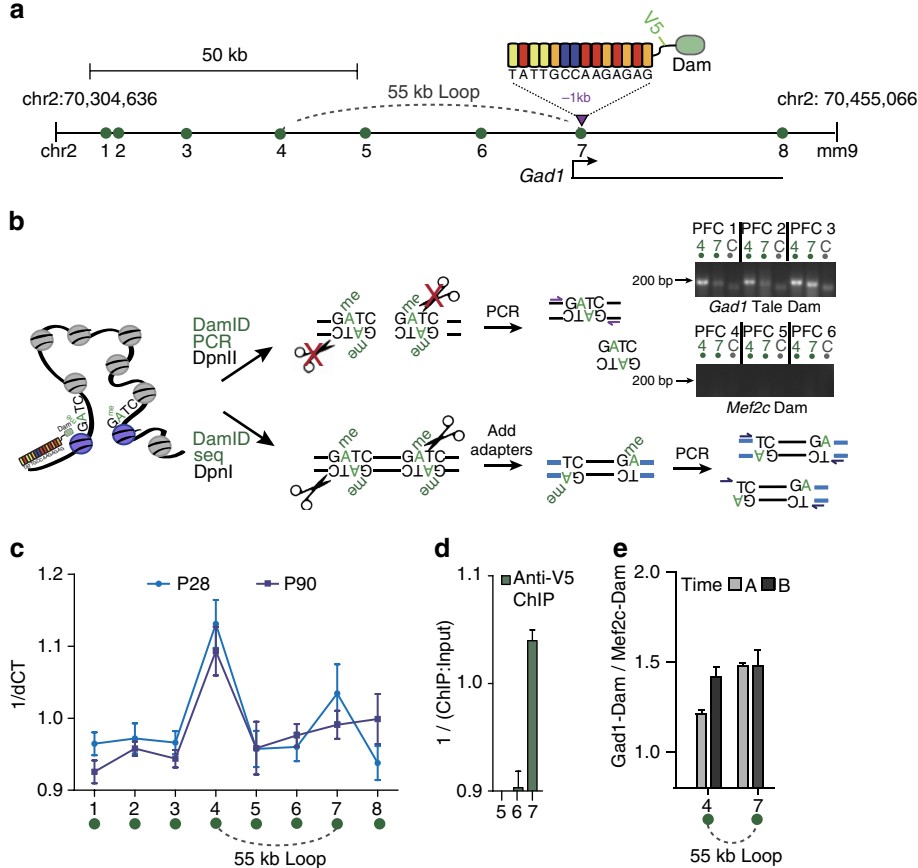

**Figure 3 | Chromosomal conformations tagged by TALE$^{Gad1}$Dam.** (**a**) 150 kb of linear genome surrounding chromosome 2 TALE target sequence 5′-TATTGCCAAGAGAG-3′ at −1 kb position from *Gad1* TSS. Dotted arc marks loop formation mapped by '3C' chromosome conformation capture[19]. Position of Amplicon/primer pairs 1–8 (Supplementary Table 2) for DamID quantitative PCR assays from DpnII-resistant prefrontal DNA as indicated within chr.2 position 70,304,636–70,455,066. (**b**) Dam-based 3D genome mapping. TALE$^{Gad1}$Dam methylates G$^{(m)}$ATC tetramers around *Gad1* TALE target sequence and at chromosomal contacts and loop formations within physical proximity to target. Methylated G$^m$ATC tetramers are selectively resistant to DpnII digest (in contrast to DpnII-sensitive non-methylated GATC). Methylated G$^m$ATC tetramers are selectively cut by DpnI (in contrast to DpnI-resistant non-methylated GATC). DamID–PCR amplifies across DpnII-resistant G$^m$ATC sequences and DamID-seq is based on adaptor-mediated ligation selectively at DpnI-sensitive G$^m$ATC. DamID–PCR products are detectable for 55 kb loop (primer pair 4), corresponding to previously reported loop formation by 3C[19] and for sequences at TALE target sequence (primer pair 7) in HSV TALE$^{Gad1}$Dam-injected PFC samples PFC1, PFC2 and PFC3. The absence of DamID–PCR product in HSV$^{Mef2c-Dam}$-injected PFC4, PFC5 and PFC6 is noteworthy (see also Supplementary Fig. 6). (**c**) DamID quantitative PCR for G$^m$ATC quantification from prefrontal DNA, with primers within 100 kb from TALE$^{Gad1}$ target sequence (see **a**), after normalization to control sequence on chromosome 18. The sharp peak at position 4, corresponding to −55 kb promoter–enhancer loop[19] and peak at position 7 at TALE target sequence are noteworthy. $N = 3$ per group. (**d**) ChIP with anti-V5 antibody to measure sequence-specific binding of TALE$^{Gad1}$Dam-V5 at *Gad1* locus. Notice robust binding at TALE target sequence (position 7, see **a**) but not at neighbouring positions 5, 6. $N = 3$ per group. (**e**) Quantitative comparison of *Gad1*-TSS$^{(−55kb)\ Loop}$ by gel densitometry from DamID–PCR products for TIME A and TIME B. $N = 4$–5 mice per group. Data in **c**–**e** shown as mean ± s.e.m.

TALE$^{Gad1}$Dam injection. Indeed, ChIP-to-Input ratios were above background at the TALE target sequence and very low or not detectable in the surrounding sequences (Fig. 3d). Importantly, DamID–PCR yields from sequences comprising the *Gad1*-TSS$^{(−55kb)\ Loop}$ were readily detectable at comparable levels in PFC harvested 2 weeks ('TIME A') and 4 months ('TIME B') after injection of HSV TALE$^{Gad1}$Dam (Fig. 3e). Furthermore, these sequences were specifically G$^m$ATC-methylated in HSV-TALE$^{Gad1}$Dam-injected PFC, whereas no DamID–PCR products were produced in 3/3 PFC specimens injected with an HSV vector expressing Dam-Mef2c transcription factor fusion protein as negative control (Fig. 3b). We conclude that G$^m$ATC tetramer methylation, as an artificial epigenetic mark reflecting bacterial Dam methyltransferase activity, is suitable for chromosomal loop mappings in mouse brain *in vivo* and, moreover, the mark is maintained in cortical neurons for at least 4 months after transient (<10day) expression.

**Dam-tagged Gad1 long range chromosomal contacts**. Next, we wanted to chart Dam G$^m$ATC-tagged sequences on a chromosome-wide scale in 'TIME A' and 'TIME B' PFC (Fig. 2). To this end, we prepared Dam-seq libraries from DpnI digested DNA to selectively ligate the adaptors at the methyl-G$^m$ATC[20] cut sites (Fig. 3b). We generated $N = 4$ TIME A PFC (pooled from four mice per library), $N = 4$ *TIME B* PFC (1 mouse per library) Dam-seq libraries, with 20 M minimum reads/library. Reads were filtered for the correct Dpn I signature (adaptor sequence followed by 5′TC…), typically comprising <5–15% of all reads. As spatial genome architectures are primarily defined by intrachromosomal contacts[21,22] and to minimize potential readings from spurious TALE binding sites elsewhere in the genome, we therefore focused our analyses on chromosome 2, which contributed 18.4% of all filtered reads ($N = 8$ PFCs Dam-seq libraries; Supplementary Table 1). This represents a several-fold genomic enrichment of chromosome 2 sequences when

corrected for chromosomal length. We also included $N = 4$ primary neuronal cultures from embryonic day E15 cortex and hippocampus transfected 72 h before harvest with TALE$^{Gad1}$Dam plasmid. Two of four primary neuronal cultures were treated with KCL for 6 h before harvest, to upregulate neuronal depolarization and signalling and thereby provide a better model for active circuitry in adult cortex. Finally, untransfected naive E15 hippocampal culture served as a negative control.

Using a 50 kb sliding window, read counts were modelled using a negative binomial distribution (see Methods). We identified 276 chromosome 2 sliding windows that were Dam-tagged in TALE$^{Gad1}$Dam-exposed PFC samples, including 57 sliding windows tagged in at least 1/4 neuronal cultures transiently transfected with TALE$^{Gad1}$Dam (Supplementary Data 1). Filter criteria included zero background in the control (non-Dam exposed) culture and exclusion of 'blacklisted' sequences in modENCODE (model Encylcopedia of DNA Elements), owing non-informative enrichment in deep-sequencing data sets[23]. The filtered 276 Dam-tagged sliding windows, in toto, represent ~7.5% of 182 megabase (Mb) mappable chromosome 2 sequence. Of note, most genomic loci exhibit a non-zero probability to interact with almost any other locus in the genome[24]. Therefore, many of the 276 Dam-tagged 'positions' on chromosome 2 could reflect 'random collisions[24]' of the chromosomal material with the TALE$^{Gad1}$ target sequence, resulting in G$^m$ATC methylation, while in physical proximity to Gad1 promoter-bound Dam enzyme. This scenario is plausible, given the longitudinal design of our in vivo experiment with the G$^m$ATC adenine methylation activity of TALE$^{Gad1}$Dam chimeric protein extending over the course of 3 days (cell culture) and 1 week (in vivo PFC). To reduce the pool of potential (Dam-tagged) intrachromosomal Gad1 contacts, we filtered for Dam-tagged sliding windows with robust ($>25$) normalized read counts, to be present in at least 3/4 TIME A plus 3/4 TIME B samples. We obtained 29/276 Dam-tagged positions that matched these criteria (Fig. 4a,b and Supplementary Data 2). Loop formations were tested by 3C for multiple candidate sequences, including formations bypassing 58 Mb, to connect Gad1 with Myo3a intronic DNA positioned next to the Gad1 orthologue Gad2 (Fig. 4c), and a long-range contact, to connect Gad1 with neurodevelopmental risk genes including Phf21a[25,26] and Kcna4 (ref. 27; Fig. 4d) and the chromatin regulator Bhc80 (refs 28,29). Additional 3C assays were conducted on an independent cohort of mice (Supplementary Fig. 3). As we verified with conventional 3C, altogether 3/4 or 75% of chromosome 2 G$^m$ATC-tagged sequences as long-range Gad1-bound intrachromosomal loopings, we conclude that the TALE$^{Gad1}$Dam fusion protein indeed left G$^m$ATC markings at regulated Gad1-bound loop formations in prefrontal neurons (as opposed to 'random collisions[24]' of the chromosomal material or spurious methylation activity of Dam not bound to the Gad1 target).

**Gad1 loop alterations mapped retrospectively.** Importantly, although the $N = 4$ TIME A and $N = 4$ TIME B Dam-seq experiments, including DNA library and sequencing, were conducted in separate batches, G$^m$ATC profiles for the 29 Dam-tagged loci nonetheless showed a moderately strong correlation between TIME A and TIME B data sets ($R = 0.64$, $r^2 = 0.4$, $P < 0.0005$). However, GenePattern-based cluster analysis using a larger set of Dam-tagged loci (Supplementary Data 1) showed that two of the altogether four TIME B Dam-seq libraries overall were poorly correlated with any other sample and therefore were not further considered (Supplementary Fig. 4A,B). Of note, the two remaining TIME B Dam-seq libraries showed robust correlations with each of the four Time A libraries (average $R = 0.815$,

which is only minimally different from the $R$ between the two Time B libraries ($R = 0.824$) (Supplementary Fig. 4A,B).

The aforementioned findings, including the comparisons between 'TIME A' and 'TIME B' samples by DamID–PCR quantification of 'local' methyl-adenine tags upstream of the Gad1 target site (Fig. 3e) and Dam-seq for intrachromosomal long-range contacts (Supplementary Fig. 4A,B) indicate that Dam-mediated G$^m$ATC profiles are maintained for at least 4 months in PFC neurons after transient Dam expression had ceased. Having shown that retrospective three-dimensional (3D) genome mapping in mouse PFC is feasible, we then wanted to explore whether our NeuroDam approach could link behavioural alterations (assessed in the 'present') to neuronal epigenome status dating back to an earlier time period in the past. To address this question, we studied our mice that were affected by changes in cognition and behaviour when tested up to 4 months after the last dose of subchronic NMDA antagonist regimen (Supplementary Fig. 1). We were particularly interested in the 58 Mb longe-range loop interconnecting the Gad1 locus with Myo3a intronic sequences positioned 30 kb upstream of Gad2 (Fig. 4a,b). Interestingly, the Gad1 and Gad2 genes encode glutamic acid decarboxylase orthologues common to all vertebrate genomes after an ancient gene duplication event $>400M$ years ago[30]. Remarkably, despite many megabases of linear genome interspersed between the Gad1 and Gad2 loci, the two genes show similar types of long-lasting adaptations in the adult mouse cortex after early life stress such as maternal immune activation. These include downregulated expression and promoter cytosine hypermethylation in context of cognitive and social impairments[31,32]. Therefore, Gad1 and Gad2 expression could show similar types of changes in our MK-801 model, in conjunction with alterations in the 58 Mb Gad1–Gad2 loop. To explore this, we quantified in the 'TIME B' PFC specimens Gad1 and Gad2 RNA (quantitative reverse transcriptase–PCR, a cross-sectional assay to quantify transcript at the time of tissue harvest) and Gad1 loopings by DamID (reflecting chromatin status at week 7, which is 15 weeks before brain harvest. Indeed, both Gad1 and Gad2 transcripts were modestly decreased (25–30%) in PFC of MK-801-exposed 'TIME B' animals Fig. 4c). This was associated together with a significant increase in Dam-tagged sequences of the Gad1-Myo3a/Gad2 loop and weakening of the local, 55 kb Gad1 looping (Fig. 4c). Therefore, long-lasting defects in cognition after MK-801 exposure are associated with early emergence of abnormal Gad1 long-range chromosomal conformations. The MK-801-induced increase in Gad1-Myo3a/Gad2 contacts was specific, because DamID for a second type of Gad1 loop structure (Gad1-TSS$^{(-55kb)\ Loop}$) showed a subtle decrease in drug-exposed animals (Fig. 4c). To explore whether Gad1 higher-order chromatin alterations persist, we conducted 3C in an additional set of 'TIMEB' PFC specimens ($N = 3$ per treatment). These 3C assays are cross-sectional, informing about the 3D genome at the time of tissue harvest or 4 months after MK-801 exposure. Indeed, 3C changes in the MK-801 group were similar to the DamID–PCR assays, which inform about 3D genome status 2–3 weeks after MK-801 exposure. Thus, our 3C–PCR studies showed a nonsignificant trend towards increased interaction frequencies for Gad1-Myo3a/Gad2 and a significant decrease in Gad1-TSS$^{(-55kb)\ Loop}$ in PFC of 'TIME B' mice previously exposed to MK-801 (Fig. 4c). Next, we directly compared in PFC from additional 'TIME A' and 'TIME B' MK-801, and saline-treated mice, the levels of G$^m$ATC adenine methylation at two disease-relevant genes, including the aforementioned Phf21a encoding a chromatin regulator associated with mental retardation and neurodevelopmental defects[25], and Kcna4 encoding a voltage-gated potassium channel broadly relevant for the regulation of neuronal excitability in the

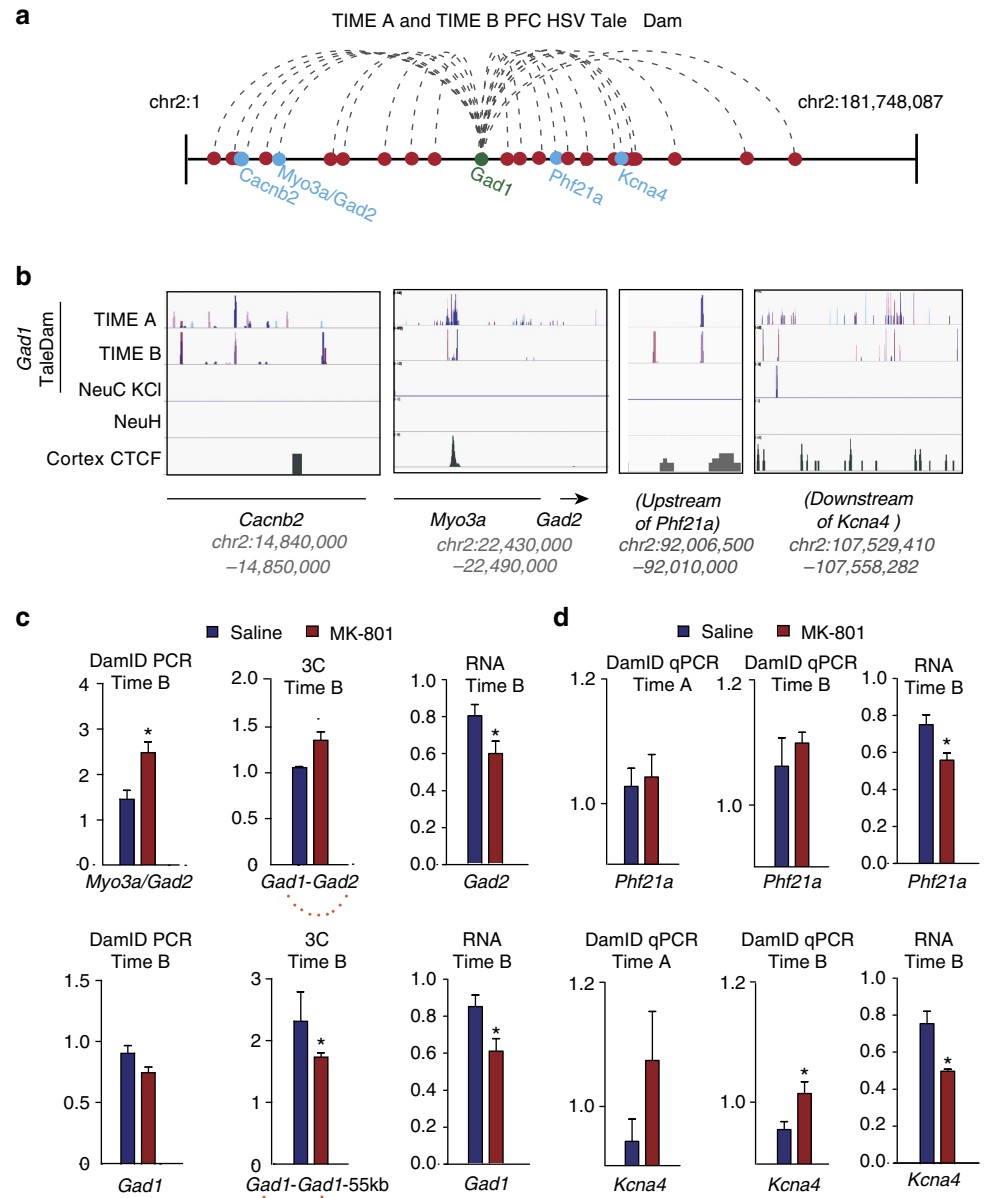

**Figure 4 | Dam-tagged long-range chromosomal loop formations in PFC neurons.** (**a**) Chromosome 2 linear map marking the positions of the 29 sliding windows consistently G^mATC-tagged in TIME A and Time B HSV TALE^Gad1Dam-injected PFC (Supplementary Data 1 and 2). (**b**) Browser view at *Cacnb2*, *Myo3a/Gad2*, *Phf21a* and *Kcna4* chromosome 2 loci (positions marked in **a**), showing normalized Dam-seq profiles (top to bottom) for $N = 4$ TIME A and $N = 4$ TIME B HSV TALE^Gad1Dam-injected PFC, $N = 2$ KCL-treated TALE^Gad1Dam primary neuronal culture. NeuH, untransfected/untreated neuronal culture. Cerebral cortex CTCF tracks built from published data set[10]. (**c**) Retrospective/longitudinal DamID and cross-sectional 3C loop assays, and quantitative reverse transcriptase—PCR (qRT–PCR) from HSV TALE^Gad1Dam-injected TIME B PFC from MK-801 and saline-treated mice for *Myo3a/Gad2*-*Gad1* long-range loop and shorter range $-55$ kb *Gad1* promoter loop. $N = 3$ per group for DamID–PCR and 3C-PCR, $N = 6$ per group for RNA. *Two-tailed *t*-test: $P = 0.015$ *Myo3a/Gad2* and $P = 0.049$ *Gad1*-TSS$^{(-55kb)}$ Loop DamID–PCR; $P = 0.058$ *Gad2* and $P = 0.034$ *Gad1* RNA. (**d**) DamID–PCR and RNA from HSV TALE^Gad1Dam-injected TIME A and TIME B PFC, for *Phf21a* and *Kcna4* sequences. DamID–PCR, $N = 3$ mice per group. qRT–PCR $N = 6$ per group *Two-tailed *t*-test: $P = 0.050$ *Kcna4* DamID–PCR, $P = 0.016$ *Kcna4* and $P = 0.027$ *Phf21a* RNA.

context of epilepsy[27]. Of note, PFC from MK-801-treated TIME A and TIME B animals showed a similar, two- to threefold increase in TALE^Gad1Dam-methylated *Kcna4*, compared with saline-treated TIME A and TIME B animals (Fig. 4d). These changes were specific, because G^mATC levels at *Phf21* showed only subtle and nonsignificant differences between treatment groups both in TIME A and TIME B animals (Fig. 4d). Unsurprisingly, expression both of *Phf21* and *Kcna4* was decreased in MK-801-exposed PFC (Fig. 4d), given the robust intra-chromosomal interactions of these gene sequences with the

*Gad1* and *Gad2* loci, which, as mentioned above, undergo a coordinated transcriptional and epigenomic response after stress[31,32].

**Dam-tagged loopings verified in additional models.** Having shown that neuronal 3D genomes in mouse PFC show long-lasting adaptations in response to NMDA antagonist drug treatment, we then tested whether chromosomal contacts discovered by Dam-seq in TALE^Gad1Dam PFC could play a role in

disease models other than MK-801. For example, prolonged social isolation (SI) in juvenile or young adult mice is associated with long-lasting changes in cognition, in conjunction with altered cortical and hippocampal glutamatergic and GABAergic circuitry[33,34]. To explore isolation-induced 3D genome adaptations, we applied 3C to hippocampi of adult mice kept either single- or group-housed from mid-adolescence (P38) onwards. Indeed, socially deprived mice showed significantly alterations in long range (Gad1-Phf21) and short range ( − 55 kb) Gad1-TSS loopings (Supplementary Fig. 3).

**Activity-dependent regulation of Gad1-bound loopings.** The experiments described above were primarily focused on Dam-based mapping of 3D genome structures in mouse brain *in vivo*, in the context of long-lasting alterations in cognition and behaviour after drug exposure earlier in life. However, previous work, using conventional 3C assays, has shown that chromosomal contacts and loopings, including Gad1-bound higher-order chromatin, could also undergo conformation changes on the scale of several hours, in the context of sudden shifts in synaptic activity[19,35]. To explore whether Dam-based chromosomal loop mapping is sensitive to such type of short-term changes in chromosomal conformations in response to neuronal depolarization, we transiently transfected primary neuronal cultures with TALE^{Gad1}Dam plasmid. Indeed, G^{m}ATC levels were significantly increased in sequences 55 kb upstream of Gad1 in depolarized (KCL-treated) neuronal cultures transfected with TALE^{Gad1}Dam plasmid (Supplementary Fig. 5), a finding that is in good agreement with the previously reported activity-dependent regulation of Gad1-bound chromosomal conformations that originate from sequences 55 kb upstream of Gad1 (Gad1-TSS^{( − 55kb) Loop} in ref. 19).

## Discussion

Here we introduce 'NeuroDam', a Dam enzyme-based technique to tag neuronal genomes in brain in longitudinal context, to link past chromatin status and present behaviours in the same animals. We expect that such type of approach will provide an important complement to the cross-sectional design characteristic of virtually all conventional neurogenomic assays available to date. Thus, NeuroDam will be a useful tool to model epigenetic mechanisms in brain disorders causally linked to risk factors operating earlier in life and leading to clinical symptoms that manifest at later points in time. For example, schizophrenia, a major psychiatric disorder with symptom onset (including altered cognition) typically in early adulthood, is considered of neurodevelopmental origin with alterations in NMDA glutamate receptor signalling negatively impacting cortical inhibitory and excitatory circuitry[36–39]. Therefore, we used NeuroDam to model schizophrenia pathophysiology in longitudinal context. We conducted a retrospective study on neuronal 3D genomes in mice that were affected by chronic changes in cognition and behaviour after a brief period of disrupted NMDA receptor signalling at juvenile or young adult age, 4 months before brain harvest ('TIME B'; Fig. 2). Although a comprehensive assessment of the molecular and behavioural changes after pharmacological NMDA receptor blockade was beyond the scope of our study, the NeuroDam findings presented here suggest that alterations in spatial genome architectures at the site of key genes implicated in the cortical dysfunction of schizophrenia, including Gad1 and Gad2 GABA synthesis genes, could occur early in the disease process (Fig. 4c,d). This finding is consistent with indirect evidence from clinical studies[40], emphasizing the potential importance of the NeuroDam approach for preclinical models of schizophrenia and related disease. Furthermore, preclinical

models of depression and posttraumatic stress disorder, in particular those that explore the genomic basis of inter-individual differences in resilience or vulnerability in the context of past exposures[41,42], are also likely to be fundamentally enriched by the retrospective genomics toolbox discussed here. Based on the findings presented here, 3D genome alterations in cortical neurons show disease-specific patterns, with a subset of chromosomal contacts, including the 58 Mb Gad1-Myo3a/Gad2 long range and the Gad1( − 55 kb) shorter range loop showing similar changes in two very different exposures (subchronic MK-801 versus SI). However, other types of interactions, including the 13 Mb Gad1-Phf21a loop, were sensitive to SI but not NMDA antagonist treatment. Therefore, it is likely that multiple, only partially interdependent regulatory networks regulate the neuronal 3D genome in locus- or sequence-specific manner.

The Dam-based tagging of chromosomal contacts *in vivo*, as shown here, will be a valuable complement to conventional restriction-ligation-based assays and microscopy-based approaches[43] to study the regulation of the 3D genome in the brain[43,44]. Furthermore, as the overwhelming majority of 3D genome contacts are intra-chromosomal, particularly in the larger chromosomes (incl. chr. 2 harbouring the Gad1 locus)[22], it remains to be determined whether NeuroDam will comprehensively inform about the 3D organization of smaller chromosomes, which tend to have a larger proportion of *trans*-chromosomal contacts. Moreover, Dam-based epigenomics in the living brain will not be confined to the study of chromosomal loopings and contacts. In non-neuronal cell cultures, Dam-based epigenomics was used to study the interactions of nuclear lamina proteins[45], distribution of linker histone subtypes[46], chromatin-remodelling complex and transcription factor occupancies[47]. As an additional benefit, transgenes encoding Dam fusion proteins can easily be engineered to limit expression to a specific cell type and/or specific period in development, in context of Cre drivers and viral vectors. This is important given that many areas of the genome are epigenetically regulated in a manner specific for cell type and developmental stage. In this context, Dam-based neurogenomics will provide an important alternative to fluorescence-activated sorting of immunotagged nuclei[48,49] or chromatin assays that rely on histone-fluorescent conjugates[43,48], or chimeric nuclear lamina proteins[50], to sort and enrich for specific cell (nuclei) types.

Of particular interest for *in vivo* studies in postmitotic cells is the apparent stability of the artificial G^{m}ATC mark, which according to our findings is readily detectable even at 4 months and longer after Dam expression (Figs 3e and 4b–d). Nonetheless, the longitudinal persistence of the Dam signal in neurons will require additional investigations. For example, regulated DNA strand breaks and DNA repair mechanisms, which could potentially 'wash out' the G^{m}ATC tag, affect neuronal genomes non-uniformly and in highly sequence-specific manner[51,52]. The absence of the G^{m}ATC tag in brain DNA not exposed to Dam (Figs 3b and 4b) could reflect the absence of adenine methylation machineries directed towards G^{m}ATC tetramers in vertebrate genomes, limiting any residual levels of adenine methylation to sequence context other than non-G^{m}ATC[15]. Furthermore, Dam-based retrospective neurogenomics may even be applicable to the invertebrate nervous system, given that adenine methylation activity in invertebrates is directed almost exclusively towards non-GATC sequences[53] conveying potentially heritable epigenetic information important for early development[53,54].

In summary, NeuroDam, as the retrospective genomics approach presented here, has the potential to illuminate the molecular bridges that link behavioural alterations in the present

to neuronal epigenome status from a distant past. Such type of approach is expected to provide a valuable alternative to conventional, cross-sectional studies that require parallel studies of multiple cohorts each harvested at a specific timepoint relative to the exposure period.

## Methods

**Animals.** All animal work was approved by the Institutional Animal Care and Use Committee of the participating institutions (Icahn School of Medicine at Mount Sinai and University of Haifa).

*MK-801 treatment.* P28 and P90 mice housed at the Icahn School of Medicine at Mount Sinai maintained in transparent cages on a standard 12:12 h light:dark cycle with *ad libitum* access to food and water were used in the study. P28 C57BL/6 mice received an initial intraperitoneal (i.p.) injection of 1.0 mg drug per kg body weight MK-801 followed by 3 weeks of a 0.2 mg MK-801 per kg body weight. Controls included saline-treated mice. All animals received saline or MK-801 injection 5 days per week. P90 C57BL/6 mice received i.p. injections of 0.2 mg drug per kg body weight saline or MK-801 for 3 weeks. Both male and female mice were used for the experiments (M:F ratio ∼10:2 for P28 and 10:4 for P90). Male and female mice were group housed with littermates ($N = 3–5$ per group).

*Adolescent SI procedure.* An independent cohort of mice, not associated with the Dam and MK-801 studies described above, was used to study the effects of SI stress. The SI procedure was performed as previously described[55]. Briefly, group-housed C57BL/6 mice were randomly assigned to group housing (GH) or SI experimental conditions on P38. Mice in the GH condition were kept in the same cage, whereas mice in the SI condition were moved to individual cages and kept in isolation for 21 days. Re-grouping in the GH condition was avoided, to minimize intruder stress in GH mice[55]. Likewise, experimenter interaction with mice from both groups was kept to a minimum (that is, cages were changed by the experimenter once a week and the experimenter was the only person in contact with the mice in both groups). All animals in the study were C57Bl/6 mice and maintained in transparent cages on a standard 12:12 h light:dark cycle in the same room with *ad libitum* access to food and water. Mice were killed on P60 and the hippocampus was dissected out bilaterally and kept at − 80 °C until further analysis by chromosome conformation capture and RNA work.

**Viral injection.** *HSV TALE^Gad1 Dam.* A TALE targeted to the Gad1 TSS (5′-TAT TGCCAAGAGAG-3′, chr2:7039932–70399346, mm9) was fused to Dam[16,56]. The Dam cDNA included a V5 epitope sequence (5′-GKPIPNPLLGLDST-3′)[56]. The Gad1-TALE-Dam was packaged into a short-term herpes simplex virus 1 with an estimated 8 day expression peaking from day 3–5 post injection driven by the IE 4/5 promoter and co-expressed with mCherry (or GFP) driven by the cytomegalovirus promoter with $4 \times 10^8$ transducing units per ml[17]. *HSV Mef2c-Dam* was generated by fusing full-length mouse Mef2c cDNA to Dam, followed by virus packaging and preparation as described above.

Animals received 2 μl of the HSV TALE^Gad1 Dam (or HSV Mef2c-Dam) virus over a period of 8 min using a syringe pump. Adult mice were anaesthetized using ketamine/xylazine (100 and 15 mg kg⁻¹, i.p.; Sigma-Aldrich, St. Louis, Missouri) mixture in PBS. A rodent stereotaxic rig mounted with a micro pump (Stoelting) and Hamilton syringe was used to bilaterally inject viral vectors into the PFC (1 μl per hemisphere). Coordinates for injection were as follows: + 1.5 mm anterior/posterior, ± 0.5 mm medial/lateral and 1.5 mm dorsal/ventral. Virus was injected per hemisphere at 0.25 μl min⁻¹ and four additional minutes were allowed before syringe removal. Control animals received 2 μl of HSV-*Mef2c-Dam* virus using the same conditions as HSV- TALE^Gad1 Dam injections.

Bilateral PFC injections (same coordinates as used for behaviour and molecular studies) in adult C57Bl/6 wild-type mice were made with HSV-Gad1-GFP and the mice were perfused 48 h post injection ($N = 4$ animals). Brains were sectioned 40 μm and screened for fluorescence pattern in serial sections starting. Of note, from four animals subjected to bilateral injections, eight of eight injections extended along the rostro-caudal axis very similar to the example provided in the new (Fig. 1c).

**Behaviour.** All behavioural experiments were carried out in the light at the onset of the animals' dark cycle by two experimenters not blinded to animal group condition.

*Eight-arm radial maze.* The maze consisted of eight arms (7.5 × 35 cm, 17.5 cm high walls) assembled radially around a circular starting platform. Mice were placed onto the starting platform and were free to enter the arms. Mice were tested until all eight arms were visited once. Each repeat entry in arm was counted as an error. Mice were trained on days 1 and 2, and tested on day 3.

*Open field.* The open-field chamber consisted of a white Plexiglas box (40 × 40 cm, 30 cm high), illuminated with bright white light (350 lux). Mice were placed individually into the box for 20 min. Total activity counts, time spent in an imaginary centre square (15 × 15 cm) of the open field and stereotypic rearing activity counts were recorded using Fusion 5.0 Superflex system.

**Immunohistochemistry and imaging.** Mice were anaesthetized with a terminal i.p. injection of a ketamine/xylazine mixture (IP: 200 and 30 mg kg⁻¹, respectively) 48 h after HSV injection. Intracardial perfusion was performed with 100 ml of 10% sucrose followed by 200 ml of 4% paraformaldehyde in PBS. Brains were removed and placed in 4% paraformaldehyde overnight at 4 °C, followed by incubation in 30% sucrose until isotonic. Brains were cut on a freezing microtome (Leica SM2010 R) into coronal sections (60 μm) and permeabilized and blocked with 0.1% Triton X-100 and 10% goat serum (Southern Biotech), respectively. Sections were incubated with NeuN-488 (1:500, EMD Millipore, ABN78A4) for 2 h, followed by a wash with PBS. Mounting was done using 4,6-diamidino-2-phenylindole Fluor-omount-G (SouthernBiotech, 0100-20). Images were taken with a Carl Zeiss CLSM780 microscope and processed using ImageJ (NIH).

**DNA/RNA extraction from PFC.** The PFC injection site harvested for the following assays: DamID–PCR and RNA expression from the same tissue samples, or DamID-seq, or conventional chromosome conformation capture (3C). DNA was isolated for DamID-seq and DamID–PCR. Tissue was homogenized in 1 Ll of $1 \times$ lysis buffer (10 mmol l⁻¹ Tris hydrogen chloride pH 8.0/10 mmol l⁻¹ sodium chloride/0.2% IPEGAL CA-630 (Sigma-Aldrich)), 5 mg proteinase K (Invitrogen) was added for 30 min at 65 °C, an equal volume of phenol:chloroform (pH 7.5) was added and the aqueous layer was taken, DNA was precipitated using 0.1 volume pH 5.2 sodium acetate and 2.5 volume 100% ice-cold ethanol overnight at − 20 °C, DNA was washed with 70% ethanol and RNaseA (Invitrogen) treated at 37 °C for 30 min. RNA was isolated for messenger RNA expression. Tissue was homogenized in 1 ml TRIzol (Invitrogen), 200 μl chloroform was added and the aqueous layer was taken, and 500 μl of isopropanol was added to precipitate RNA overnight at − 20 °C.

**Dam-seq and DamID–PCR.** DamID-seq libraries were prepared using DpnI. Ten micrograms of DNA was cut at 37 °C overnight using 10 units DpnI (New England Biolabs, Boston, Massachusetts). AdR adapters were exposed to boiling water and allowing the adapters to pair as the water slowly came to room temperature. Adapters (40 nmol) were ligated to DpnI-cut DNA at 16 °C for 2 h using 5 units T4 DNA ligase (Invitrogen). Non-DpnI-restricted DNA was cut with DpnII (New England Biolabs) for 1 h at 37 °C. Adapter-ligated DpnI-cut DNA was amplified using PCR primers. DamID–PCR libraries were created using DpnII. Five micrograms of DNA was cut at 37 °C overnight using 5 units of DpnII (New England Biolabs). Primers were designed across DpnII restriction sites and DNA amplified by PCR.

Libraries were 100 bp end sequenced on the Illumina HiSeq 2000. Quality was assessed using FASTQ and had an average quality score of 34. Reads were sorted based on the Gad1 DpnII sequencing primer and trimmed by the primer and adapter length of 42 bp. Single end reads were aligned to mm9 using Burrows-Wheeler Aligner (BWA) and duplicates removed using SAMTOOLS. Chromosome 2 data were analysed in 50 kb sliding windows using a negative binomial (gamma Poisson) distribution. To normalize for within-group variability, samples were modelled by a dispersion parameter. Genes of similar expression levels were assumed to have similar dispersion and dispersion was estimated using maximum likelihood for separate loci and smooth curve was fit. Gene-wise dispersions were shrunk towards values predicated by the smooth curve using empirical Bayes[57]. In addition, sequences considered by the ENCODE and modENCODE consortia, via manual curation and automated heuristics, as prone to artefact signal in next-gen sequencing (ChIP sequencing, MNase sequencing, DNase sequencing and formaldehyde-assisted isolation of regulatory elements (FAIRE) sequencing) assays[23] were removed from analyses.

To compare Time A and Time B Dam-seq adenine methylation profiles, data were normalized using the negative binomial distribution. Loci with normalized read counts > 10 (Supplementary Data 1) were used in Pearson's hierarchical clustering by genomic loci and sample using GenePattern (www.broadinstitute.org) pairwise average-linkage with mean-based row and column centring[58].

DamID–PCR. Samples were amplified using $1 \times$ GoTaq Green master mix (Promega). PCR cycling conditions were 95 °C for 5 min, 40 cycles of 95 °C for 30 s, 60 °C for 30 s and 72 °C for 30 s, one cycle of 95 °C for 30 s, 60 °C for 30 s and 72 °C for 8 min and a final hold at 4 °C. Signal intensity was measured via ImageJ or SYBR green-based quantitative PCR. Control PCR reactions, to ensure complete digestion at GATC tetramers, were designed with primers around DpnII sites at the transcription start site of Cxxc5, a transcription factor on chromosome 18 with no Gad1 interactions.

All relevant data are available from the authors.

**Chromosome conformation capture.** Rostro-medial cortex was homogenized and cross-linked for 10 min at 25 °C in 1% formaldehyde, $1 \times$ protease inhibitor (Sigma) and 2 ml lysis buffer (10 mmol l⁻¹ Tris hydrogen chloride pH 8.0/ 10 mmol l⁻¹ sodium chloride/0.2% IPEGAL CA-630 (Sigma Aldrich)). A final concentration of 0.125 mol l⁻¹ glycine was added for 10 min to stop cross-linking. The homogenate was incubated for another 25 min at 4 °C. Cells were lysed by pipetting > 50 times and spun at 5,000 r.p.m. Supernatant was removed and the pellet was washed twice with $1 \times$ New England Buffer 4 (NEB4) (New England Biolabs). Samples were resuspended in 200 μl of $1 \times$ NEB4 and divided into four

50 µl aliquots. An additional 312 µl of $1\times$ NEB4 and 38 µl of 1% SDS were added to each aliquot and the samples were incubated at 65 °C for 10 min. To quench the SDS, 10% of Triton X-100 was added to each sample and the samples were digested with HindIII-HF (NEB) at 37 °C overnight with vigorous shaking. HindIII-HF was inactivated by the addition 86 µl of 10% SDS incubated for 30 min at 65 °C. Ligation mixture (7.61 ml) was added to each sample. The ligation mixture consisted of 745 µl of 10% Triton X-100, 745 µl of $10\times$ ligation buffer (1 M Tris HCl pH 7.5, 1 M MgCl$_2$, 1 M dithiothreitol (Bio-Rad)), 80 µl of 10 mg ml$^{-1}$ BSA (NEB), 80 µl of 100 mM ATP (Sigma) and 5,960 µl of autoclaved water. Fifty microlitres of T4 DNA ligase (1 U µl$^{-1}$, Invitrogen) was added to three aliquots and one sample was used as a no ligase control. Ligation proceeded for 5 h at 16 °C and samples were reverse cross-linked at 65 °C overnight with 50 µl of 10 mg ml$^{-1}$ of proteinase K (Sigma). For improved ligated DNA recovery, another 50 µl of proteinase K was added and incubated at 65 °C for 2 h. DNA was extracted with phenol (pH 8.0, Fisher) and phenol–chloroform (1:1) (pH 8, Fisher). DNA was precipitated using 1/10 the volume of 3 M sodium acetate (pH 5.4) and 2.5 the volume of ice-cold ethanol overnight. The samples were centrifuged at 8,000 r.p.m. for 30 min and washed with 70% ethanol. The final DNA pellet was dissolved in $1\times$ TE buffer (pH 8.0). Phenol and phenol–chloroform extraction and ethanol precipitation was repeated. The final 3C library was washed five times with 70% ethanol. Ligase and no ligase reactions were dissolved in 100 and 33 µl of TE buffer (pH 8.0) respectively[59]. Ligase and no ligases libraries alone were run on a 2% agarose gel to visualize ligation efficiency. Samples ran at a higher molecular weight after ligation, indicated by an upward shift on the gels[60,61].

Physical interactions between non-contiguous sequences were quantified using PCR. Samples were PCR amplified using $1\times$ GoTaq Green master mix (Promega). PCR cycling conditions were 95 °C for 5 min, 40 cycles of 95 °C for 30 s, 60 °C for 30 s and 72 °C for 30 s, one cycle of 95 °C for 30 s, 60 °C for 30 s and 72 °C for 8 min and a final hold at 4 °C. Primers were designed less than 120 bp from a HindIII restriction site. The PCR products were resolved on a 2% agarose gel and the level of interaction between two primers was measured semiquantitatively using band intensities normalized with the background (raw 3C interaction) with ImageJ[62]. Library input was adjusted for each library according to the interaction between two neighbouring primers ($<$5,000 bp apart) and two distant primers ($<$30,000 bp apart): control primer 1 (5′-CCTGG ATCATCAGACAGAACTAAAGCTCTT-3′) located at chr13:99113854 and control primer 2 (5′-CTTCAACTGAAAACACACGAACAGGAAGAA-3′) located at chr13:99109553. Specificity of 3C PCR products was confirmed by sequencing. Furthermore, 3C assays as described above, but with no ligase added, and water served as negative controls. 3C assays were normalized to a neighbouring primer (Gad1n, Supplementary Table 2).

**ChIP with anti-V5 antibody.** ChIP analysis was performed using HSV-TALE$^{Gad1}$-Dam(V5)-injected PFC samples from $N=4$ mice. Tissue was dounced in 1 ml lysis buffer (10 mmol l$^{-1}$ Tris hydrogen chloride pH 8.0/10 mmol l$^{-1}$ sodium chloride/0.2% IPEGAL CA-630 (Sigma-Aldrich)) and 1% formaldehyde for 10 min. A final concentration of 0.125 mol l$^{-1}$ glycine was added for 10 min, to stop cross-linking. Homogenate was washed in sonication buffer (50 mM Hepes pH 7.9, 140 mM NaCl, 1 mM EDTA, 1% Triton X-100, 0.1% Na-deoxycholate and 0.1% SDS). Samples were sonicated on high 30 s on and 30 s off for 30 min. Pierce Protein A/G Magnetic Beads (Thermo-Scientific Fisher, 88802) were washed in sonication buffer. Samples were precleared in 30 µl A/G beads for 1 h at 4 °C. Ten per cent of the sample was collected after preclearing for use as input. Primary anti-V5 (Thermo-Scientific Fisher, R96025) was added overnight at 4 °C. A/G beads were added samples for 3 h at 4 °C. Samples were washed at 4 °C twice with sonication buffer for 5 min, twice with wash buffer A (50 mM Hepes pH 7.9, 140 mM NaCl, 1 mM EDTA, 1% Triton X-100, 0.1% Na-deoxycholate, 0.1% SDS and 500 mM NaCl) for 5 min, twice with wash buffer B (20 mM Tris pH 8, 1 mM EDTA, 250 mM LiCl, 0.5% NP-40 and 0.5% Na-deoxycholate), twice with $1\times$ TE buffer (10 mM Tris pH 8.0 and 1 mM EDTA) for 5 min and eluted with elution buffer (50 mM Tris pH 8, 1 mM EDTA and 1% SDS) for 5 min at 65 °C and 10 min at room temperature. DNA was precipitated from input and ChIP samples using an overnight 65 °C treatment of 5 M NaCl, followed by a 2 h proteinase K treatment at 65 °C and phenol–chloroform (pH 8) extraction. DNA was precipitated from the aqueous layer using 1/10 volume sodium acetate (pH 5.2) and 2.5 volume 200 proof ethanol.

**mRNA expression.** Prefrontal mRNA expression was quantified from cDNA produced from 1 µg RNA using the ABI High Capacity cDNA kit (Applied Biosystems). RNA expression was quantified using Power SYBR green master mix (Life Technologies) for 40 cycles. Quantitative PCR cycling conditions were one hold of 95 °C for 10 min, 40 cycles of 95 °C for 15 s and 60 °C for 1 min, and a denaturation step. *Dam*, *Gad1* and *Gad2* quantifications were expressed after normalization to 18S ribosomal RNA.

**Primary cultures.** Embryonic day 15 (E15) cortical and hippocampal tissue was dissected and dispersed for primary cultures. Six wells (9.5 cm$^2$ per well) of primary cultures were transiently transfected with TALE$^{Gad1}$Dam plasmid or mock transfected using NeuroFect (Amsbio) and $1\times10^6$ cells for each condition were harvested after 72 h.

**Data availability.** All relevant data, protocols and reagents will be available from the authors.

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

## Acknowledgements

We thank Dr Bart van Steensel for the generous Dam cDNA gift and for providing technical advice. This study was supported by grants from the National Institutes of Health. We thank Dr Bisrat Woldemichael and Brian Safaie for critically reading the manuscript.

## Author contributions

A.C.M. and S.A. designed the project. A.C.M. contributed to all aspects of the study. A.C.M., B.J. and L.K.B. performed the experiments. R.N., Y.J., K.G., K.M., S.S.L. and I.G.-S. provided reagents and materials. H.M., M.A.W. and S.A. provided resources.

## Additional information

**Competing financial interests:** The authors declare no competing financial interests.

