## [Peer Review File · Nature Communications]

Reviewers' comments:

Reviewer #1 (Remarks to the Author):

Mitchell et al used a Dam enzyme based technique, 'NeuroDam', to tag neuronal genomes and attempted to assess epigenome status in longitudinal context. They used the NMDA-sensitive Gad1 GABA synthesis gene as a target locus, which was artificially tagged by a transiently expressed Dam. They then measured the animal's behavior, and assessed prefrontal genomes at different time points. By performing Dam-seq, they identified Gad1-bound long range chromosomal contacts, and found that chronic deficits induced by disrupted NMDA receptor signaling in the juvenile and young adult period, is associated with long-lasting disruptions of intrachromosomal conformations. Generally, the study is well-designed and well-done, I believe that the 'NeuroDam' will provide a powerful tool in the field of epigenetic regulation. I would therefore recommend publication of the manuscript in NC after minor revision.

Minor comments:

1. In the figure 3D (left), please add scale bars to these panels (Mef2c).
2. The authors showed that they identified five GmATC-methylation sequences in HSV-TALEGad1 Dam/mCherry injected PFC, but it is not clear the number of specimens was used in the experiment.
3. In page 6, the authors treated mouse with saline or MK-801, yet they showed GmATC levels were increased of Gad1 in depolarized. Furthermore, are changes in GmATC levels in saline or MK-801 treated mice?
4. In page10, "Indeed, both Gad1 and Gad2 transcripts were decreased (30-40%) in Dam-tagged sequences of the Gad1-Myo3a/Gad2 loop" should refer to Figure 5A and B.
5. Could the authors comment as to why DamID for a second type of Gad1 loop structure showed no significant MK-801 effects?
6. In figure 1B, the authors also should show scale bar in overlay panel.
7. The authors showed that both Gad1 and Gad2 transcripts were decreased (30-40%) in PFC of 'TIME B' MK-801 animals, but in Figure 5A and 5B, it seems that there was no difference in statistical analysis about the expression of Gad1 and Gad2.

Reviewer #2 (Remarks to the Author):

Summary:

The main strength of the manuscript entitled "Longitudinal Assessment of Neuronal 3D Genomes in Mouse Prefrontal Cortex" is the introduction of a novel NGS-based technique, referred to as NeuroDam, in neuropsychiatric research. This technique (first used to investigate the bithorax complex in *Drosophila*) might allow the retrospective analysis of cis-regulatory interactions at a given locus. The expression of a chimeric protein composed of a TALE DNA-binding domain fused to a bacterial DAM domain should allow the deposition of methyl-GmATC in sequences that interact with the sequence targeted by the TALE construct. This idea is intriguing, innovative and has great potential. However, it is not obvious which is the advantage provided by NeuroDam in the context of the experiments presented in this manuscript. Unfortunately, the experiments presented in the article do not demonstrate the utility of the NeuroDam approach nor conclusively show that it works with the expected level of specificity (important controls are missing). Furthermore, the organization of text and figures makes difficult to follow the motivation of some experiments.

Main

1. As a proof-of-concept, the authors investigated the appearance of conformational changes in the chromatin associated with chronic blockade of NMDA receptor signaling. It is known that chronic exposure to the NMDA-antagonist MK801 has enduring effects in working memory and anxiety in mice. It is surprising that the authors decided to place all the information regarding the characterization of this model as Supplementary material when it is key for demonstrating the biological relevance of their study and to support important statements in the text. The reason for this decision might be that the behavioral experiments did not provide the expected results. According to the data shown in Supplemental Fig 1, the authors did not fully reproduce some previous findings in this paradigm. MK801-treated young mice (P28 group) showed no significant difference in working memory at Time B when compared with mice treated with saline (in fact, learning for all groups was very poor and there

were almost no differences in performance (ratio of errors) upon training). In the case of older mice (P29 group), the treatment did not produce significant differences in anxiety neither at Time A nor B. In addition, when presenting the data, the authors should consider that repeated measures increases the likelihood of type I errors and post-hoc comparisons are not corrected for such phenomenon. The weakness of the behavioral alterations reduces the biological relevance of all the study. What are the implications of the reported genomic alterations after MK801 for brain function and behavior?

2. Figure 3D indicates that repeated injections of MK801 alter some chromosomal interactions as detected by NeuroDam. It is surprising that the authors did not explore further the consequences of these changes in the expression of the locus and in Gad1 activity. Figure 5B indicates that the levels of Gad1 are not significantly affected. As indicated above, it remains unclear what are the implications of the reported genomic alterations after MK801 for brain and cellular function.

3. In the manuscript the authors tagged chromosomal interactions during a given period (referred to as Time A) and investigated whether this tagging still persisted at a later time point (Time B). This is not a longitudinal assessment of 3D interactions. The way in which the technology is employed does not provide any answer that could not be obtained using traditional 3C assays. In order to make the longitudinal assessment the authors could investigate whether the chromosomal interaction occurring at time B (present) are the same ones tagged during Time A (past). Then, they could demonstrate the coincidence of long-lasting aberrant chromosomal interactions with long-lasting behavioral deficits.

4. The correlation presented in Fig 4C is very low ($r^2 = 0.4$). This value does not conclusively support the maintenance of the marks. Fig 4B and the inset in 4C also indicates that there are very important differences between Time-A and time-B which could argue against the use of NeuroDam in longitudinal studies.

5. The discovery-driven part of the study (Fig. 4) is underdeveloped. Why did the authors focus exclusively in chromosome 2 interactions? They are overlooking a large part of the information contained in their datasets.

6. Organization: The main experiment is introduced in Fig 2 (a scheme presenting the experimental design that should precede Fig 1) and its results are presented in Sup Fig 1, Fig 1, Fig 3D, Fig 4 and Fig 5A-B. This presentation is interrupted by the NeuroDam experiments in primary cultures (Fig 3C) with no obvious relevance in the context of the study. The experiment in isolated animals presented in Fig. 5C is also poorly justified and explained. These two experiments also use NeuroDam but it is not clear how they support the main study. Rather than evaluate the technique in three different, but poorly characterized settings, the authors could dedicate more detail to demonstrate that the NeuroDam technique works as expected using a single paradigm.

Specific points:

7. Abstract. "Retrospective profiling of neuronal epigenomes is likely to illuminate epigenetic determinants of normal and diseased human brain development in longitudinal context". How are the authors planning to apply NeuroDam to humans? It does not seem possible.

8. Figure 1A. Scale makes difficult to see the results for Gad1 and Gad2.

9. Figure 1B. There is no description of the reproducibility, efficiency and extent of the viral infection across animals.

10. How specific is the binding of the TALE construct? Can the authors use ChIP assay or (better) ChIPseq to demonstrate that it binds exclusively or primarily to the target sequence at the Gad1 locus?

11. What is "Control" in Figure 3C? Is not explained in the figure legend. How was selected the sequence in chr18?

12. Figure 3D. All the explored regions around Gad1 present the same behavior (presence of loops enabling GACT methylation). These values do not support the existence of a specific interaction between region 5 (target of TALE construct) and any other region. There is no internal negative

control (intra-locus or close to Gad1) for the Dam methylation assay. I would also recommend that the impact of MK801 in the interactions were presented as absolute values rather than ratios.

REVIEWERS' COMMENTS:

Reviewer #1 (Remarks to the Author):

The authors have addressed most of my concerns. I would like to recommend publication of the revised manuscript on NC.

Reviewer #2 (Remarks to the Author):

The revised version of the manuscript added a number of experiments, including several critical controls, that strengthened the study and supported the validity of this proof-of-concept study. Furthermore, the addition of new text and the reorganization of some paragraphs and figure panels streamlined the presentation of the results. Overall, the novel version is very much improved.

Minor comment:

- Line 185, page 8 indicates "...to connect Gad1 with two chromatin regulators and neurodevelopmental risk genes Phf21a25 and Bhc8027 (Supplemental Figure 3)". However Supp. Fig. 3 refers to an independent experiment on isolated animals. The authors must be referring to panels that are not included in the current version of the manuscript. Also, the description of Supp. Fig. 4 preceded that of Supp. Fig. 3 (after correcting the mistake commented above) and their order should be altered.

Point-by-point response:

Introductory Remarks and Overview:

We very much appreciate the constructive comments provided by the two Reviewers and the Editor. We now substantially revised the paper, and addressed each of the Reviewers' comments. Newly added experiments and analyses included in the revised paper are listed here, and a detailed point-by-point response to the previous Review included further below.

Newly added experiments and analyses:

1. Profiling of G^mATC methylation by DamID PCR at the *Gad1* locus, now with additional primer pairs and controls (revised *Figure 3C*).
2. Chromatin immunoprecipitation with anti-V5 (epitope tag) antibody to show selective enrichment of the TALE-Dam(V5) fusion protein at the TALE target sequence (revised *Figure 3D*).
3. Additional quantitative comparisons of G^mATC methyl-adenine levels at long range *Gad1* chromosomal contacts in drug- and saline-treated mice, with two different time timepoints for each treatment group (revised *Figure 4D*).
4. Additional immunohistochemical studies in prefrontal cortex of HSV TALE^{Gad1}-Dam injected mice and additional documentation on region-specific delivery of the expression vector (revised *Figure 1*).
5. Additional quantitative measurements of RNA expression in MK-801 and saline-treated animals (revised *Figure 4C,D*), now providing a stronger link between chromosomal contacts and co-regulated gene expression.
6. Additional correlational analyses of adenine methylation profiles from brains harvested at different time points (*Supplemental Figure 4*).

Detailed point-by-point response:

Reviewer #1

1. *In the figure 3D (left), please add scale bars to these panels (Mef2c).*

Response: We thank the Reviewer for making us aware of this oversight. We now added the bp/size standard to both gels (revised *Figure 3B* (=former figure 3D)).

2. *The authors showed that they identified five GmATC-methylation sequences in HSV-TALE^{Gad1} Dam/mCherry injected PFC, but it is not clear the number of specimens was used in the experiment.*

Response: We appreciate this comment and now have added the information into the figure legend (revised figure 3C) (N=3-group). We also increased the number of sequences tested from previously 5 to now 8, at the *Gad1* locus.

3. *In page 6, the authors treated mouse with saline or MK-801, yet they showed GmATC levels were increased of Gad1 in depolarized. Furthermore, are changes in GmATC levels in saline or MK-801 treated mice?*

Response: We appreciate this comment and have in the revised version, moved the description of the ex vivo depolarization experiments towards the end of the Results section, to provide better separation from the paper's main focus (in vivo Dam tagging). We conducted additional experiments and quantified GmATC levels for specific genes both for Time A and B treated MK-801 vs. saline mice (revised *Figure 4C, D*).

4. *In page10, "Indeed, both Gad1 and Gad2 transcripts were decreased (30-40%) in Dam-tagged sequences of the Gad1-Myo3a/Gad2 loop" should refer to Figure 5A and B.*

Response: We revised the order and numbering of the figure and now refer to the Gad1 and Gad2 work to revised *Figure 4C*.

5. *Could the authors comment as to why DamID for a second type of Gad1 loop structure showed no significant MK-801 effects?*

Response: We followed the Reviewer's suggestion and now added additional text to the Discussion section (second page) to emphasize the emerging view of the neuronal 3D genome, with disease- and locus-specific patterns.

6. *In figure 1B, the authors also should show scale bar in overlay panel.*

Response: We appreciate this comment. In response, we revised Figure 1, by adding additional panels (now showing histological sections from prefrontal cortex at day 2 and day 10 post-injection, at three different magnifications). We added scale bars for each magnification and added the info into the figure legend accordingly.

7. *The authors showed that both Gad1 and Gad2 transcripts were decreased (30-40%) in PFC of 'TIME B' MK-801 animals, but in Figure 5A and 5B, it seems that there was no difference in statistical analysis about the expression of Gad1 and Gad2.*

Response: We appreciate this comment and have, since the original submission, conducted additional qRT-PCR studies to increase the N. The new results are provided in Figure 4C, with the difference between MK-801 and saline now reaching significance. This is now indicated in the figure and in the figure legend.

Reviewer #2

1. *It is surprising that the authors decided to place all the information regarding the characterization of this model as Supplementary material when it is key for demonstrating the biological relevance of their study and to support important statements in the text. The reason for this decision might be that the behavioral experiments did not provide the expected results. According to the data shown in Supplemental Fig 1, the authors did not fully reproduce some previous findings in this paradigm. MK801-treated young mice (P28 group) showed no significant difference in working memory at Time B when compared with mice treated with saline (in fact, learning for all groups was very poor and there were almost no differences in performance (ratio of errors) upon training). In the case of older mice (P29 group), the treatment did not produce significant differences in anxiety neither at Time A nor B. In addition, when presenting the data, the authors should consider that repeated measures increases the likelihood of type I errors and post-hoc comparisons are not corrected for such phenomenon. The weakness of the behavioral alterations reduces the biological relevance of all the study. What are the implications of the reported genomic alterations after MK801 for brain function and behavior?*

Response: In response to this comment, we now emphasize more clearly, in the first paragraph of the discussion section, that a comprehensive behavioral assessment of the long-term consequences of subchronic NMDA receptor blockade would be beyond the scope of this study. The main focus of this paper is on the Dam-based neuronal genome tagging technique in vivo. In this regard, our paper marks a true first in the field. Therefore, we feel it is fully appropriate to keep the MK-801 behavioral data in the supplemental data section. Moreover, to the best of our knowledge, the long-term behavioral consequences after subchronic MK-801 exposure are still incompletely understood. Importantly, the large majority of previous studies covered shorter periods as compared to the 4 months applied here. Therefore, the Reviewer's statement '*...the behavioral experiments did not provide the expected results*' is not correct.

2. *Figure 3D indicates that repeated injections of MK801 alter some chromosomal interactions as detected by NeuroDam. It is surprising that the authors did not explore further the consequences of these changes in the expression of the locus and in Gad1 activity. Figure 5B indicates that the levels of Gad1 are not significantly affected. As indicated above, it remains unclear what are the implications of the reported genomic alterations after MK801 for brain and cellular function.*

Response: We appreciate this comment and have, since the original submission, conducted additional qRT-PCR studies to increase the N. The new results are provided in Figure 4C, with the difference between MK-801 and saline now reaching significance. This is now indicated in the figure

and in the figure legend. We decided, based on the Reviewer's comment, to pursue additional RNA quantifications and tested two additional genes (*Kcna4* and *Phf21a*) that are loop-bound with *Gad1*. Indeed, both genes show also decreased expression in MK801 treated animals. These newly obtained results are presented in the revised Figure 4D. These additional data are described in the last paragraph of the Results section subchapter. We write 'Unsurprisingly, expression both of *Phf21* and *Kcna4* was decreased in MK-801 exposed PFC (Figure 4D), reflecting the intrachromosomal interaction of these gene sequences with the *Gad1* and *Gad2* loci which, as mentioned above, undergo a coordinated transcriptional and epigenomic response after stress (referenced Labouesse et al. 2015, Richetto et al., 2014).'

Finally, we would like to reiterate to the Reviewer that the primary focus of our paper is to introduce a novel, innovative strategy to map, retrospectively, the neuronal 3D genome. Any in-depth exploration of the neurobiology of MK-801 exposed brain will be deferred to future studies.

- In the manuscript the authors tagged chromosomal interactions during a given period (referred to as Time A) and investigated whether this tagging still persisted at a later time point (Time B). This is not a longitudinal assessment of 3D interactions. The way in which the technology is employed does not provide any answer that could not be obtained using traditional 3C assays. In order to make the longitudinal assessment the authors could investigate whether the chromosomal interaction occurring at time B (present) are the same ones tagged during Time A (past). Then, they could demonstrate the coincidence of long-lasting aberrant chromosomal interactions with long-lasting behavioral deficits.*

Response: We appreciate this comment. However, as clearly outlined in the timeline (Figure 2), we tagged chromosomal interactions prior to TIME A and TIME B brain harvest. We studied the 3D genome at multiple timepoints, hence the use of the word 'longitudinal' is justified: We mapped retrospectively, the 3D genome in brains in TIME A and TIME B animals, in addition to the cross-sectional approach informing about the 3D genome at the time of tissue harvest (conventional chromosome conformation capture, 3C). We included NeuroDam data from TIME A and TIME B animals, in addition to 3C (Figure 4C,D).

While the original submission had included chromosomal loop comparative quantification both for TIME A and TIME B, the previously shown data were limited to the 50-70kb of sequence surrounding the *Gad1* locus. Therefore, in response to the Reviewer's comment, we decided to include additional comparisons. We quantified, by DamID PCR, G^mATC levels at additional TALE^{Gad1}-Dam target sites, including sequences at *Kcna4* and *Phf21*, for TIME A and TIME B MK-801 and saline-treated mice. These additional experiments are now presented in Figure 4D.

- The correlation presented in Fig 4C is very low ($r^2 = 0.4$). This value does not conclusively support the maintenance of the marks. Fig 4B and the inset in 4C also indicates that there are very important differences between Time-A and time-B which could argue against the use of NeuroDam in longitudinal studies.*

Response: We thank the Reviewer for this important comment. We would like to point out that an $r^2=0.4$, which equals $R=0.64$, is generally considered a moderately strong correlation, but certainly not a 'very low' correlation as the Reviewer claims. Nonetheless, in response to the Reviewer's comment, we conducted additional correlational analyses using Pearson hierarchical clustering by genomic loci and sample, as now described in the Methods section (subchapter *Dam-seq and DamID-PCR*). The new results are presented in *Supplemental Figure 4A,B*. Note that all Time A libraries were processed together in a single batch, separately from Time B libraries that also were processed together in a single batch separate from TIME A. Essentially, 2 of the 4 TIME B Dam-seq libraries showed robust correlations with each of the four Time A libraries (average $R=0.815$), which is only minimally different from the R among (these two) Time B libraries (average $R=0.824$). One additional TIME B library showed a much weaker correlation, and the 4th TIME B library showed negative correlations the other libraries (*Supplemental Figure 4A,B*), and therefore of questionable quality. Taken together, these findings strongly suggest that good quality Dam-seq libraries show strong correlations among brains harvested at different time points after transient

vector (Dam) expression. These additional analyses and findings are now described in the Results section.

5. *The discovery-driven part of the study (Fig. 4) is underdeveloped. Why did the authors focus exclusively in chromosome 2 interactions? They are overlooking a large part of the information contained in their datasets.*

Response: We appreciate this comment but would like to emphasize that the major focus of our paper is to introduce a new method to the field—retrospective 3D genome mapping in mouse brain in vivo. Because the overwhelming majority of 3D genome contacts are intra-chromosomal, we feel that the focus on chromosome 2 (which harbors the *Gad1* locus) makes sense for our paper. Stringent exploration of trans-chromosomal contacts will require additional validation and control experiments and we would like to defer those to future studies. In response to the Reviewer's comment, we now have added additional into the Discussion section (2nd paragraph), emphasizing that the overwhelming majority of chromosomal contacts, particularly for large chromosomes (including chromosome 2 harboring the *Gad1* locus) are intra-chromosomal, and that future NeuroDam studies in some of the smaller chromosomes (which tend to have a larger proportion of trans-chromosomal contacts) will explore the potential of our technique to comprehensively chart trans-chromosomal 3D genome contacts.

6. *Organization: The main experiment is introduced in Fig 2 (a scheme presenting the experimental design that should precede Fig 1) and its results are presented in Sup Fig 1, Fig 1, Fig 3D, Fig 4 and Fig 5A-B. This presentation is interrupted by the NeuroDam experiments in primary cultures (Fig 3C) with no obvious relevance in the context of the study. The experiment in isolated animals presented in Fig. 5C is also poorly justified and explained. These two experiments also use NeuroDam but it is not clear how they support the main study. Rather than evaluate the technique in three different, but poorly characterized settings, the authors could dedicate more detail to demonstrate that the NeuroDam technique works as expected using a single paradigm.*

Response: We appreciate this comment. We agree with the Reviewer that the flow of the in vivo work presentation is unnecessarily interrupted by the cell culture work, and hence we moved the cell culture figure panel now to a new Supplemental figure (*Supplemental Figure 5*) and moreover, we describe the culture work now at the end of the results section. We would like to include these cell culture data in the paper because previous work (cited in the paper) documented that some of the chromosomal loop formations undergo dynamic changes in cultured cells. In the current paper, we are able to show that the NeuroDam approach, applied to cell culture, provides similar results as previously reported with conventional chromosome conformation capture assays.

Furthermore, to accommodate the second point raised by the Reviewer in the comment 'Organization', we now moved the figure summarizing the social isolation experiment to the Supplemental Figure section (revised Supplemental Figure 3), and discuss this experiment only after we described the in vivo NeuroDam work. We now provide additional explanation for the social isolation study (the experiment confirms that chromosomal loop formations discovered by NeuroDam could be verified in different mouse colony and in a different disease model and an alternative technique (3C). These findings would suggest that the approaches introduced by our paper could be broadly applied in various contexts and disease paradigms.

7. *Abstract. "Retrospective profiling of neuronal epigenomes is likely to illuminate epigenetic determinants of normal and diseased human brain development in longitudinal context". How are the authors planning to apply NeuroDam to humans? It does not seem possible.*

Response: We apologize for this confusion and removed the word 'human' from the abstract.

8. *Figure 1A. Scale makes difficult to see the results for Gad1 and Gad2.*

Response: We appreciate this comment and now included an additional Supplemental Figure (*Supplemental Figure 2*) to better visualize the results for *Gad1* and *Gad2*.

9. *Figure 1B. There is no description of the reproducibility, efficiency and extent of the viral infection across animals.*

Response: We appreciate this comment. To address this issue, we have added two additional panels to Figure 1, directly comparing HSV vector-mediated transgene expression at day 2 and day 10 (we previously only showed day 2) and a low power image of a bilateral injection and serial coronal sections on which a representative injection site is charted. We also provide additional details in the methods section, stating that in order to test for consistency among injections, serial coronal sections were prepared from 4 animals with bilateral injections and essentially 8/8 injections extended along the rostro-caudal axis very similar to the example provided in the new Figure 1C.

10. *How specific is the binding of the TALE construct? Can the authors use ChIP assay or (better) ChIPseq to demonstrate that it binds exclusively or primarily to the target sequence at the Gad1 locus?*

Response: We appreciate this excellent comment. We conducted now additional experiments, taking advantage of the fact that Bart Stensel's laboratory already had fused a V5 epitope tag to the Dam cDNA. We run ChIP from PFC (two days after vector injection) using a V5 antibody. These newly gained results are now included in the revised figure 3, panel D. A subchapter has been added to the Methods section ('*Chromatin immunoprecipitation with anti-V5 antibody*'), describing the anti-V5 ChIP procedure. There was specific enrichment of (Dam)V5 occupancy at the TALE target sequence, while signal was very low at sequence upstream of the target sequence; this is consistent with the very low level adenine methylation levels at these sequences.

11. *What is "Control" in Figure 3C? Is not explained in the figure legend. How was selected the sequence in chr18?*

Response: We appreciate this comment. We now specify in the Methods subchapter 'DamID-seq and DamID-PCR' the Dam control PCR with primers to amplify a chromosome 18 sequence.

12. *Figure 3D. All the explored regions around Gad1 present the same behavior (presence of loops enabling GACT methylation). These values do not support the existence of a specific interaction between region 5 (target of TALE construct) and any other region. There is no internal negative control (intra-locus or close to Gad1) for the Dam methylation assay. I would also recommend that the impact of MK801 in the interactions were presented as absolute values rather than ratios.*

Response: We appreciate this comment. To address these issues, which relate to our DamID PCR experiment in sequences surrounding the *Gad1* locus, we now conducted a new set of experiments, increasing the number of sequences tested from (previously) 5 to now 8, across a wider window (formerly 50kb, now 100kb), and refined the quantification method of the PCR, using qPCR. The new data are presented in the revised Figure 3A,C, now clearly show in two independent cohorts ('TIME A' P28 and P60 mice that had been exposed to HSV-Gad1-Dam, a highly specific interaction between the target of the TALE construct and sequences positioned 55kb further upstream (PCR primer pair no. 4 in the revised figure).

Response to Reviewer #1:

The authors have addressed most of my concerns. I would like to recommend publication of the revised manuscript on NC. Response: Thank you.

Response to Reviewer #2:

Minor comment: - Line 185, page 8 indicates "...to connect Gad1 with two chromatin regulators and neurodevelopmental risk genes Phf21a25 and Bhc8027 (Supplemental Figure 3)". However Supp. Fig. 3 refers to an independent experiment on isolated animals. The authors must be referring to panels that are not included in the current version of the manuscript. Also, the description of Supp. Fig. 4 preceded that of Supp. Fig. 3 (after correcting the mistake commented above) and their order should be altered.

Response: We appreciate this comment and in response, have revised portions of the last paragraph of the Results subchapter 'Gad1-bound long range chromosomal contacts identified by Dam-seq'. Specifically, we now state clearly that these 3C verification studies were conducted in multiple independent cohorts of mice (including the animals that provided for the data shown in Supplementary Figure 3). The supplemental figure numbering remains correct.